# Polyols and Polyurethane Foams Based on Water-Soluble Chitosan

**DOI:** 10.3390/polym15061488

**Published:** 2023-03-16

**Authors:** Anna Maria Strzałka, Jacek Lubczak

**Affiliations:** 1Doctoral School of Engineering and Technical Sciences, Rzeszow University of Technology, Al. Powstańców Warszawy 6, 35-959 Rzeszów, Poland; 2Faculty of Chemistry, Rzeszów University of Technology, Al. Powstańców Warszawy 6, 35-959 Rzeszów, Poland

**Keywords:** water-soluble chitosan, hydroxyalkylation, polyols, polyurethane foams

## Abstract

At present, majority of polyols used in the synthesis of polyurethane foams are of petrochemical origin. The decreasing availability of crude oil imposes the necessity to convert other naturally existing resources, such as plant oils, carbohydrates, starch, or cellulose, as substrates for polyols. Within these natural resources, chitosan is a promising candidate. In this paper, we have attempted to use biopolymeric chitosan to obtain polyols and rigid polyurethane foams. Four methods of polyol synthesis from water-soluble chitosan functionalized by reactions of hydroxyalkylation with glycidol and ethylene carbonate with variable environment were elaborated. The chitosan-derived polyols can be obtained in water in the presence of glycerol or in no-solvent conditions. The products were characterized by IR, ^1^H-NMR, and MALDI-TOF methods. Their properties, such as density, viscosity, surface tension, and hydroxyl numbers, were determined. Polyurethane foams were obtained from hydroxyalkylated chitosan. The foaming of hydroxyalkylated chitosan with 4,4′-diphenylmethane diisocyanate, water, and triethylamine as catalysts was optimized. The four types of foams obtained were characterized by physical parameters such as apparent density, water uptake, dimension stability, thermal conductivity coefficient, compressive strength, and heat resistance at 150 and 175 °C. It has been found that the obtained materials had most of the properties similar to those of classic rigid polyurethane foams, except for an increased thermal resistance up to 175 °C. The chitosan-based polyols and polyurethane foams obtained from them are biodegradable: the polyol is completely biodegraded, while the PUF obtained thereof is 52% biodegradable within 28 days in the soil biodegradation oxygen demand test.

## 1. Introduction

At present, most polyurethane foams (PUF) are obtained from petrochemical substrates [1,2,3]. The fact that petroleum resourcesare decreasing makes it necessary to look for resources of a biological origin. Polyols based on renewable natural resources are easily available, comparatively not expensive, and biodegradable. The constant increase of foam products on one hand and environmental requirements on the other hand imposes on the searchfor biodegradable polyols from plant oils [4,5] or carbohydrates such asstarch and cellulose [6,7,8,9,10,11]. Chitosan also belongs to the latter group of compounds and was not used untilnow to obtain polyols. Chitosan can be obtained by D-deacetylation of chitin (Figure 1). Chitin is apolymer that is present in marine invertebrates and in the skeleton material in crustaceans and insects [12].

Chitosan can be considered as a natural resource because this polymer is the building material of the fungal cell wall. It can be easily isolated by extraction. Chitosans are a group of polymers of variable degrees of molecular weight and deacetylation. These physicochemical characteristics of fungal chitosan such as molecular weight and degree of degradation can be better controlled compared to chitosan obtained from crustacean sources [13]. Fungal chitosan has many advantages in biomedical applications due to its molecular properties. It is easier to obtain very low or high molecular mass polymer from fungal chitosan compared to shellfish chitosan. Fungal chitosan can be used as a potential carrier of drug and non-viral gene delivery systems [13].

The chitosan chain comprises the units of β-(1,4)-D-glucoseamine and N-acetyl-D-glucoseamine (Figure 2) [14]. The distributions of deacetylated and acetylated subunits within the polymeric chain influence the chemical properties of chitosan. There are various kinds of chitosan commercially available, which differ in molecular weight and deacetylation degree. Generally, chitosan is a biodegradable, biocompatible, and non-toxic polymer with some antigen properties [15]. It is used in tissue engineering and as an ingredient of drug delivery systems, diet supplements, cosmetics, in plant cultivation, and in environmental technologies [16,17,18,19,20], and also to separate dyes and heavy-metal ions [21,22].

Chitosan used in industry is not soluble in water, although it dissolves in aqueous solutions of organic acids such as formic, acetic, and citric at pH levels below 6.5 [12]. In order to convert chitosan, the amine groups at C-2 and/or primary and secondary hydroxyl groups at C-4 and C-6, respectively, can be used. Thus O- and N-functionalized derivatives with carboxymethyl, acyl, sulfone, and alkyl derivatives were obtained as well as coordination compounds with metal ions [23]. Due to the lack of systemic toxicity, biocompatibility, and facile biodegradation, many derivatives of these polymers have found their application in medicine, tissue engineering, and pharmacy [24,25,26,27]. In recent years, chitosan-based material shave found a further application in skin tissue engineering [28]. Furthermore, the chitosans with free amine groups are applicable in wastewater treatment [24]; for example, binary coagulation system graphene oxide/chitosan was used for polluted surface water treatment [29,30].

Amine and hydroxyl groups of chitosans provide a path to obtain polyols suitable for further use to obtain polyurethane foams (PUF) in a similar way as starch or cellulose derivatives [31,32]. Chitosan-derived polyols were not reported, except for in [33], although hydroxyalkylation of chitosan methods were elaborated [33,34,35]. Additionally, chitosan-filled PUFs were described. Thus, the hydroxyalkylation of chitosan in 15% aqueous NaOH with ethylene, propylene, and butylene oxides, and then further reaction with oxirane, led to a hydroxyalkylated derivative of chitosan. This product was grafted on collagen or nisin to obtain the sorbent materials suitable for pharmaceutic and medical applications [35,36]. Similar hydroxyalkylation with glycidol was also reported [37]. Generally, the chitosan hydroxyalkylated with epoxides can be further substituted with carboxyl groups [38,39]. The application of chitosan to obtain polyols was proposed by Fernandes et al. [33]. They have activated chitin and chitosan with potassium hydroxide and treated them with propylene oxide to obtain polyol, which was suitable for further conversion to polyurethanes and polyesters. However, the obtained polyols had high viscosity, which rendered them immiscible with diisocyanate to obtain PUF. They were also contaminated with the catalyst. As mentioned earlier, the chitosan-derived polyols were not used as a component to obtain PUFs, although it has been shown that composites of PUF and chitosan were an elastomeric product. The PUFs with added chitosan were also biodegradable, which promised their application as sorbents and biomaterials [40,41,42,43,44,45,46,47].

In this paper, for the first time, chitosan was used to obtain polyols and rigid polyurethane foams. Four methods of polyol synthesis were elaborated, namely by hydroxyalkylation of water-soluble chitosan with glycidol and ethylene carbonate in water and in glycerol, as well as without a solvent. The new series of rigid polyurethane foams were obtained, which are susceptible for biodegradation.

## 2. Materials and Methods

### 2.1. Materials

The following materials were used in this work: water-soluble chitosan, degree of deacetylation, DD = 85.8%, viscosity molecular weight, *Mv*~14, 1250 Da (CS, Biosynth-Carbosynth, Staad, Switzerland), glycidol (GL, pure 98%, Sigma-Aldrich, Taufkirchen, Germany), ethylene carbonate (EC, pure ≥ 99%, Fluka, Buchs, Switzerland), potassium carbonate (anal. grade 100%, POCH, Gliwice, Poland), polymeric diphenylmethane 4,4′–diisocyanate (pMDI, Merck, Darmstadt, Germany), triethylamine (TEA, anal. grade ≥ 99%, Fluka, Buchs, Switzerland), surfactant Silicon L-6900 (pure, Momentive, Wilton, CT, USA), and glycerol (GLYC, anal. grade 99.5–100%, POCH, Gliwice, Poland).

### 2.2. Synthesis of Polyols

#### 2.2.1. Synthesis 1: Polyol (CS + H_2_O + GL) + EC

Here, 6 g of CS, 60 g of GL, and 45 g of water were placed in a three-necked round-bottom flask equipped with a reflux condenser, mechanical stirrer, and a thermometer. The mixture was heated at 105–110 °C until complete reaction of GL (determined by the epoxide number). Then, water was distilled under reduced pressure (p = 30 mm Hg, up to temperature 150 °C). The product was a clear gelatinous liquid (the semi-product was CS + H_2_O + GL). To this semi-product, EC (75 g) and 0.5 g of potassium carbonate (catalyst) were added. The mixture was reheated up to 160 °C and maintained until all the EC was reacted.

#### 2.2.2. Synthesis 2: Polyol (CS + GLYC + GL) + EC

Here, 6 g of CS, 60 g of GL, and 45 g of GLYC were placed in a three-necked round-bottom flask equipped with a reflux condenser, mechanical stirrer, and a thermometer. The mixture was heated at 160 °C until complete reaction of GL. Afterward, to this semi-product (CS + GL + GLYC), EC (105 g) and potassium carbonate (0.5 g) were added, and the mixture was heated at 180 °C until EC was completely consumed.

#### 2.2.3. Synthesis 3: Polyol (CS + GL) + EC

CS (6 g) and GL (90 g) were placed in a three-necked round-bottom flask, equipped with a reflux condenser, mechanical stirrer, and a thermometer. The mixture was heated at 190 °C until complete reaction of GL. Then, to the semi-product (CS + GL), EC (135 g) and potassium carbonate (0.5 g) were added, and the mixture was heated at 170 °C.

#### 2.2.4. Synthesis 4: Polyol (CS + GL + EC)

CS (6 g), GL (75 g), and EC (90 g) were placed in the flask and heated at 140–145 °C, until chitosan was dissolved. Then, the temperature was increased to 160 °C and the mixture was refluxed. During stepwise consumption of GL, the temperature increased to 190 °C and the mixture was maintained at this temperature for ca 0.5 h until GL was fully consumed. Then, the mixture was cooled down to 140 °C, potassium carbonate (1 g) was added, and the mixture was kept at 140 °C until EC was completely reacted.

### 2.3. Analytical Methods

The deacetylation degree of chitosans was determined according to the results of the elemental analysis, as it was described in [48]. Molecular mass was determined by the viscosimetric method at 30 °C using the Mark–Houvink equation, as described in [49]:[η] = k M_υ_^α^(1)
where: [η] is intrinsic viscosity, M_υ_ is the viscosity-average mass weight, and k, α are constants that are characteristic for a particular polymer–solvent system at a specific temperature:k = (1.64 · 10^−30^) · (DD^14^) [cm^3^/g](2)
α = (−1.02 · 10^−2^) · (DD) +1.82(3)
where DD is the % degree of deacetylation.

The reaction of the mixture of CS with GL was monitored by epoxide number determination using hydrochloric acid in dioxane [50]. Specifically, 25 cm^3^ of hydrochloric acid solution in dioxane (1.6 cm^3^ in 100 cm^3^ dioxane)was added into a 0.5 g mass sample. Excess of HCl was then titrated with 0.2 M NaOH in methanol in the presence of *o*-cresol red as an indicator. The progress of the reaction of hydroxyalkylation with EC was monitored using the barium hydroxide method described in [51]. In particular, the samples of 0.1–0.5 g of mass were treated with 2.5 cm^3^ of 0.15 M Ba(OH)_2_ and then titrated with 0.1 M HCl in the presence of 0.2% thymoloftalein in alcohol. Finally, the hydroxyl number (HN) of polyol was determined by acylation with acetate anhydride in dimethylformamide [52]. Thus, 1 g of sample was heated with a 20 cm^3^ acetylating mixture (acetic anhydride and dimethylformamide at a 23:77 v:v ratio) for 1 h at 100 °C. Excess of anhydride was titrated with 1.5 M NaOH_aq_ in the presence of phenolphthalein. The ^1^H-NMR spectra of reagents were recorded using a500 MHz Bruker UltraShield instrument in DMSO-d_6_ and D_2_O with hexamethyldisiloxane as an internal standard. IR spectra were registered on an ALPHA FT-IR BRUKER spectrometer in KBr pellets or by the ATR technique. The samples were scanned 25 times, in the range from 4000 to 450 cm^−1^ at a 2 cm^−1^ resolution. MALDI-TOF (Matrix-Associated Laser Desorption Ionization Time of Flight) spectra of polyols were obtained on a Voyager-Elite Perceptive Biosystems (US) mass spectrometer working in linear mode with delayed ion extraction, equipped with a nitrogen laser working at 352 nm. The method of laser desorption from gold nanoparticles (AuNPET LDI MS) was applied [53]. The observed peaks corresponded to the molecular K^+^ (from catalyst) ions. The samples were diluted with methanol to 0.5 mg/cm^3^.

### 2.4. Physical Properties of Polyol

Density, viscosity, and surface tension of polyol were determined with a pycnometer, Höppler viscometer (type BHZ, Prüfgeratewerk, Germany), and by the detaching ring method, respectively.

### 2.5. Polyurethane Foams

Foaming of polyol was performed in500 cm^3^ cups at room temperature. The foams were prepared from 10 g of polyol, to which 0.30–0.39 g of surfactant (Silicon L-6900) and 0.08–0.27 g of TEA as a catalyst and water (2–3%) as a blowing agent were added. After homogenization, the polymeric diphenylmethane 4,4′-diisocyanate was added in the amount of 11.0–18.5 g. The commercial isocyanate containing 30% of tri-functional isocyanates was used. The mixture was vigorously stirred until creaming began. The materials were then conditioned at room temperature for 3 days. The samples for further studies were cut from the obtained foam.

### 2.6. Properties of Foams

The apparent density [54], water absorption [55], dimensional stability at 150 °C [56], thermal conductivity coefficient (IZOMET 2104, Bratislava, Slovakia), and compressive strength [57] of PUF were measured. The apparent density of PUFs was calculated as the ratio of PUF mass to the measured volume of the PUF sample in a cube of a 50 mm edge length. Water volume absorption was measured on cubic samples of 30 mm edge lengths by full immersion of PUF in water and mass measurement after 5 min, 3 h, and 24 h. Dimensional stability was tested on samples of 100 mm × 100 mm × 25 mm in size. The thermal conductivity coefficient was measured at 20 °C after 72 h of PUF conditioning. The needle was inserted 8 cm deep into a cylindrical PUF sample 8 cm in diameter and 9 cm high. Compressive strength was determined using burden causing 10% compression of PUF height related to the initial height (in accordance with the PUF growing direction). The thermal resistance of modified foams was determined by both static and dynamic methods. In the static method, the foams were heated at 150 and 175 °C with continuous measurement of mass loss and determination of mechanical properties before and after heat exposure. The 100 × 100 × 100 mm cubic samples were used to determine the static thermal resistance and compressive strength. In the dynamic method, thermal analyses of foams were performed in a ceramic crucible in a20–600 °C temperature range, with about 100 mg of sample, under air atmosphere with a Thermobalance TGA/DSC 1 derivatograph, Mettler, with a 10 °C/min heating rate. Topological pictures of PUFs were recorded for cross-sections of PUF samples. The pictures were analyzed with a Panthera microscope (prod. Motic, Wetzlar, Germany) with 4 × /0.13, 10 × /0.30 lenses and worked up with Motic Multi-Focus Professional 1.0 software, enabling merging and manipulation of images with adjustable lensing planes.

### 2.7. Biodegradation of Polyol and Foam

The biodegradation of polyol and the PUF obtained from it was tested by the use of the OxiTop Control S6 instrument (WTW-Xylem, Rye Brook, NY, USA). The respirometric method was used to measure the oxygen demand necessary for aerobic biodegradation of polymeric materials in soil. The measurement of consumed oxygen was presented using the value of biochemical oxygen demand (*BOD*), which is the number of milligrams of captured oxygen per mass unit of the tested polyurethane material. The instrument was composed of six 510 cm^3^ glass bottles, equipped with rubber quiversand measuring heads, which were used to measure the *BOD*. They allowed to measure the pressure in the range of 500 to 1350 hPa with an accuracy of 1% at a temperature of 5 to 50 °C. The communication between the measuring heads and the user was performed with Achat OC computer software (WTW-Xylem, Rye Brook, NY, USA), which was applied to interpret the obtained measurement results.

The biodegradation tests were performed according to the norm [58]. For a biodegradation test, the sieved and dried gardening soil was used with the following parameters: 5% humidity (according to ISO 11274-2019 [59]), pH = 6 (according to ISO 10390-2005 [60]), and particle diameters < 2 nm. The measurement was carried out in a system consisting of 200 mg of the tested sample (oligomer or foam), 200 g of soil, and 100 g of distilled water. The samples were homogenized in bottled, rubber quivers containing two pastilles of solid NaOH, and were mounted and sealed with measuring heads for six samples. The set was incubated at 20 ± 0.2 °C for 28 days. The current oxygen consumption was determined within 2–3-day intervals for the samples and 2 references: positive and negative, plus a blank, which was the soil and water only. The starch was used as the positive sample, while polyethylene was the negative sample. BOD was determined for every sample, taking into account the *BOD* of the tested system reduced by the BOD of the soil and the concentration of the tested compound in the soil using the following formula:(4)$$BODs=\frac{{BOD}_{x}-{BOD}_{g}}{c}$$
where: *S*—number of measurements (in days), *BOD_S_*—biochemical oxygen demand of the analyzed sample within S days (mg/dm^3^), *BODx*—biochemical oxygen demand of the measuring system (bottle with sample and soil) (mg/dm^3^), *BODg*—biochemical oxygen demand of the soil without a sample (mg/dm^3^), and c—sample concentration in the tested system (mg/dm^3^). The degree of biodegradation of the polyol or the foam based on it was determined using the formula:(5)$$D_{t}=\frac{{BOD}_{S}}{TOD}\cdot 100\%$$
where *Dt* is the biodegradation degree of the sample (%) and *TOD* is the theoretical oxygen demand (mg/dm^3^).

The theoretical oxygen demand was calculated using the formula provided in norm ISO17556-2019 [59]. It has been assumed that in oxygen conditions, the carbon is converted into CO_2_, hydrogen into H_2_O, and nitrogen into NH_3_.

For the compounds of known *C, H, N,* and *O* percentages and total mass of the sample, the *TOD* value can be calculated from the following equation:(6)$$TOD=\frac{16\cdot [2C+0.5\cdot \left(H-3N\right)-O]}{m}$$
where *C, H, N,* and *O* are the mass fractions of elements in the biodegraded material, and *m* is the sample mass of the material (g).

## 3. Results and Discussion

### 3.1. Obtaining of Polyols

The known methods of CS hydroxyalkylations with oxiranes are difficult to perform. Oxiranes are low boiling, toxic, and flammable liquids, cancerogenic, and form an explosive mixture with air. Therefore, using them requires high-pressure reactors. Common ways to convert CS require preliminary treatment of CS with NaOH in alcohol, followed by a reaction with an oxirane [33]. Side products are formed in the reaction of alcohols with alkylene oxides. Thus, in order to use CS as a substrate for PUF, it needs to obtain a semi-product: a liquid polyol suitable to react with diisocyanate. Chitosans of high molecular weight were not good candidates for such a conversion because of their low solubility. Therefore, the water-soluble CS was chosen for a reaction with GL. Our earlier experience [10,11] on hydroxyalkylation of starch and cellulose showed that those sparingly soluble polymers could be successfully hydroxyalkylated by preliminary heating the substrates in water with GL, which enabled to obtain a better soluble substrate which was further converted by a reaction with EC to obtain polyols. We applied the elaborated method to convert the CS in water with GL at a slightly elevated temperature (40 °C). CS itself is a product of chitin deacetylation in which the degree of deacetylation (DD) is 85%. That rendered the CS water-soluble and enabled to convert it in the reaction with GL. The reaction mixture was then gradually heated up to reflux (ca. 100 °C). The product analysis by determination of the epoxide number indicated that GL was consumed in the reaction with water to yield GLYC (Figure 3).

The amount of water distilled from the reaction mixture was 26.4 mass % instead of theoretically 27.4 mass % if all the GL were to react with water. Thus, this semi-product contained GLYC, which was isolated after the initial removal of water, and identified by IR, refraction index, and MALDI-TOF of polyol (vide infra). We concluded that unreacted and dissolved CS was present in the post-reaction mixture. The obtained semi-product was semisolid resin, not miscible with isocyanates, and was then further liquefied by hydroxyalkylation with EC according to Figure 4.

The amount of EC was minimized in order to obtain polyol of low viscosity, miscible with diisocyanate, pMDI. Preliminary experiments of the direct reaction between CS and EC in the presence of the K_2_CO_3_ catalyst at 180 °C led to carbonization of the polymer. We also found that water was not a necessary solvent to dissolve CS, and GLYC could be used as a solvent for CS and hydroxyalkylation could be performed with GL, and further with EC. Thus, the way to obtain the polyol could be simplified due to avoiding water removal. A general scheme of chitosan hydroxyalkylation is presented in Figure 5. This remains valid for other methods of polyol synthesis further described below.

Obtained liquid polyols contain side products, namely the products of the reaction between glycerol and glycidol (see Table 1), for example those in Figure 6.

Further attempts indicated that the synthetic pathway could be simplified by straightforward hydroxyalkylation of CS with excess GL, without using GLYC. The obtained semi-product required consecutive hydroxyalkylation with EC in order to obtain a final product of low viscosity. 

Finally, we performed the one-pot synthesis of polyol by introducing CS, GL, and EC into a reaction flask. We have previously found that CS does not react with EC without a catalyst. Thus, the EC acted as a solvent for CS, while GL reacted with CS. After consuming all GL, the catalytic amount of K_2_CO_3_ was added to trigger the reaction with EC.

All tested methods resulted in the formation of liquid polyols, miscible with pMDI.

### 3.2. Composition and Structure of Polyols

The progress of the reaction was monitored by spectroscopes IR and ^1^H-NMR and the MALDI-TOF technique. The spectra of obtained polyols were compared with those of the starting CS (Figure 1). The IR spectrum of CS showed a broad band centered at 3400 cm^−1^from hydroxyl and amine groups’ stretching vibrations, as well as deformation bands at 1420 cm^−1^ and 1630 cm^−1^, respectively. The band centered at ca. 1030 cm^−1^ was attributed to valence of ether C-O-C vibrations. The presence of acetylamine groups was demonstrated by I and II amide bands at 1630 cm^−1^ and 1520 cm^−1^ (overlapped with the amine deformation band of chitosan). The IR spectra of the obtained polyols are presented in Figure 2. The IR spectra of all polyols were similar due to the similarity of the chemical structural fragments. The increase in intensity of the C-O-C band at 1030 cm^−1^ was observed as well as methine and methylene bands (2900 cm^−1^, 1400–1300 cm^−1^), which derive from glycidol ring opening and the incorporation into polyol. The presence of a carbonyl band at 1750 cm^−1^ indicated that ester bonds are present in polyol, especially those obtain data temperature lower than 180 °C. In such conditions, carbonate groups are able to incorporate into the polyol structure, as it was observed before in [51].

In the ^1^H-NMR spectra of CS (Figure 3), the amine group protons yielded resonances at ca. 8.2 ppm, while primary and secondary hydroxyl group protons were present at 4.8 ppm and 5.5 ppm. Between the latter, the characteristic resonance of C_1_H from a chitosan ring was present. The C_3_-C_6_ methine proton resonance multiplets were present at 3.5–3.9 ppm regions, while the C_2_H signal overlapped with a water residual broad signal. Methyl resonance from acetylamine was observed at 1.9 ppm [14]. The ^1^H-NMR-obtained polyols are shown in Figure 4. In these spectra, the amine proton resonances disappeared due to hydroxyalkylation of amine groups.

The region 3.2–3.5 ppm was considerably modified, and the additional resonances from methylene and methine protons appeared, which evidenced the GL and EC ring opening and incorporation of their structural fragments into the polyol. The chitosan hydroxyl resonances (previously observed at 4.8 and 5.5 ppm) disappeared, while new hydroxyl proton resonances grew within the 4.3–4.6 ppm region. No considerable differences were observed between polyol obtained from various chitosans and variable conditions of the reaction. Using mass spectrometry in the MALDI-TOF technique, the side products formed in polyol syntheses were identified. The illustrative example of the results is shown in Table 1 for (CS + GL) + EC polyol obtained by method 3. Low molecular weight peaks corresponded to not-reacted GL (Table 1, entries 1, 2). There were also the peaks corresponding to the product of hydroxyalkylation of GL and its oligomers with EC with elimination of CO_2_ (Table 1, e.g., entries 4, 5, 7–9, 11, 14, 18, and Figure 7 and Figure 8).

Side products can also be dehydrated in the reaction conditions (Table 1, for example entries 10, 15–17, and Figure 9).

The MALDI-TOF spectrum of polyol (CS + GL + H_2_O) + EC obtained in water confirmed the presence of GLYC, which was formed in the reaction between water and GL (Table 2, entry 5). It can be further hydroxyalkylated with GL (Table 2, entries 9, 13, 14, 21, 26). The obtained oligomers can then react with EC (Table 2, entries 11, 15, 17–20, 22–25). The MALDI-TOF spectrum of polyol (CS + GL + GLYC) + EC contained similar peaks as the previous case because GLYC was added into the reacting system (Table 3). In both spectra, the products of oligomerization of GL and the GL + EC reaction were present, similarly to the spectrum of the aforementioned polyol (CS + GL) + EC.

The obtained polyols are liquids miscible with pMDI. Their hydroxyl numbers and physical properties, such as density, viscosity, and surface tension, were determined. The results are collected in Table 4. The temperature dependencies of the physical parameters of the studied polyols are typical of those used to produce PUFs (Figure 5) [61]. Polyols obtained in water or in GLYC showed a lower density and viscosity, which were caused by the presence of a reactive solvent in the system. The hydroxyl numbers within 409–654 mg KOH/g suggested that the obtained polyols can be used to obtain rigid PUFs. The high values of hydroxyl numbers of polyols (CS + GL + H_2_O) + EC and (CS + GL + GLYC) + EC were due to the presence of products of water and glycerol hydroxyalkylation (high functionality, high number of hydroxyl groups).

### 3.3. Preparation of Polyurethane Foams

The attempts to use the obtained polyols as substrates for the synthesis of PUFs were performed on a laboratory scale in order to select and optimize the kind of polyol, the amount of diisocyanate (pMDI), a catalyst, a surfactant, and a foaming agent. We aimed at a rigid PUF with small pores (Table 5). We concluded that the most promising PUFs were obtained when the amount of pMDI corresponding to the molar ratio of the isocyanate group to the hydroxyl group (isocyanate index, II) was within 1.1–1.3. The exemption was PUF obtained from the (CS + GL + EC) polyol, which was obtained with II 1.5. The relatively high value of II can be attributed to a higher share of chitosan units in the polyol structure, and thus a higher number of amino groups. Amine groups can catalyze trimerization of isocyanates to isocyanuric rings, and this side-reaction may result in isocyanate consumption. Thus, the high thermal resistance of PUF obtained from this polyol can be caused by the high thermal resistance of isocyanuric rings [62]. The optimized surfactant Silicon L-6900 amount was 3.0 or 3.9 g per 100 g of polyol. Two PUFs were obtained from every polyol by using a variable amount of the foaming agent (water), i.e., 2% and 3% related to the polyol mass. The PUFs obtained with less than 2% of water were under-foamed, while those obtained with more than 3% of water were semi-rigid with large pores. The amount of TEA used was variable within 0.5–2.7 g/100 g of polyol. The optimized amount of the catalyst depended on the water content and decreased when the water was increased. The lowest amount of catalyst was successfully used for compositions obtained from (CS + GL + GLYC) + EC and (CS + GL) + EC polyol. When less than the optimized amount of TEA was used, the PUF had irregular, large pores, and was under-crosslinked, while when more than the optimal amount of TEA was used, fast growth and a limited size of PUF was observed, which finally led to a lower foaming degree and an increase of the apparent density of PUF. Creaming, rising, and tack-free times were observed during foaming. They depended on the kind of polyol and the amount of catalyst. The creaming time for optimized compositions was within 21–72 s, while the rising time was short (22–59 s). The longest rising time was observed for the composition with (CS + GL) + EC polyol and 3% water. A typical increase of the creaming time and the rising time of the compositions due to a decreased amount of the catalyst was noticed. The tack-free time was very short (below 6 s) for compositions which used polyols synthesized with water and GLYC, and a longer time (above 10 s) for other PUFs.

### 3.4. Properties of Polyurethane Foams

The following properties of obtained PUFs were determined: apparent density, water absorption (by volume), dimensional stability, thermal conductivity coefficient, thermal resistance, compressive strength, and glass transition temperature. The apparent density of the obtained PUFs is illustrated in Figure 6. The PUFs obtained with a lower amount of the foaming agent had a higher apparent density due to lower foaming. Thus, the largest density was observed for PUFs obtained from compositions with 2% water (70–73 kg/m^3^).

Water absorption in the 24 h test was low (usually below 3%), suggesting that the closed pores dominated in the obtained PUFs (Figure 7a,b). This low water absorption is caused by the presence of open cells on the cut of PUF. In order to visualize the pores, the images of cross-sections of PUFs were taken.

Statistical analysis of images enabled to determine an average pore size and thickness of the pore walls (Figure 8). It can be concluded that oval pores of various sizes were present (Table 6). Since it was an ellipsoidal shape, two diameters were calculated. The average longer diameter was within 169–321 µm, while the shorter one was within 84–152 µm. Longer pores were observed in PUFs obtained from (CS + GL + H_2_O) + EC and (CS + GL) + EC polyols if 3% water was present in the foaming compositions (Table 6). This led to the release of more CO_2_ and also caused a decrease of thickness of the pore walls. Small pores in PUF obtained from (CS + GL + GLYC) + EC polyol and 2% water/100 g of polyol led to the highest compressive strength of all the obtained PUFs. The larger pores enabled to absorb water, and thus the PUFs with large pores showed higher water absorption, especially in cases of PUFs obtained from (CS + GL + H_2_O) + EC and (CS + GL + GLYC) + EC polyols.

Thermal conductivity coefficient values of the obtained PUFs (Figure 9) in the presence of 2% water were similar to those found in typical rigid PUFs (0.0260 W/m·K) [61].

The obtained PUFs had a good dimensional stability at elevated temperatures, which fell to −2.80% to 3.0% after a 40 h exposure at 150 °C (Table 7). In some cases, elongation in one and shortening in another dimension of exposed PUFs has been observed.

Thermal resistance of PUFs for one month of exposure at 150 °C and 175 °C was determined by the mass loss. The mass loss was due to physical changes, such as water diffusion and evaporation and chemical conversions. The PUFs were not resistant to thermal exposure at 200 °C already in the first days of exposure. The mass loss upon thermal exposure is illustrated in Figure 10a,b. The largest mass loss was observed in the first days of thermal exposure because of the initial water and catalyst (TEA) removal. Thus, after one month of exposure at 150 °C, the mass loss was within 10.1–17.4% (Table 8). After one month at this temperature, the lowest mass loss was observed for the PUFs obtained from (CS + GL + H2O) + EC polyol and the polyol from one-pot synthesis, namely (CS + GL + EC).

These two PUFs also had high thermal resistance at 175 °C. High thermal resistance in the first case is related to high crosslinking of PUF obtained from polyol (CS + GL + H_2_O) + EC, which showed a high value of the hydroxyl number as well as the highest percentage of thermally resistant chitosan units in this polyol (5.1%), in comparison to polyols from syntheses 2(3.6%) and 3 (3.7%). Chitosan decomposes at ca. 280 °C [63]. Another polyol containing a slightly lower proportion of chitosan units (4.8%) is (CS + GL + EC). The highest mass loss was found for PUFs obtained from (CS + GLYC + GL) + EC polyol and the 2% foaming agent (17.4% and 34.6% at 150 °C and 175 °C, respectively). The PUFs at ambient temperature were rigid and thermal exposure at 150 °C and 175 °C did not change this.

The obtained PUFs are characterized by compressive strength typical of classic, rigid polyurethane foams (Table 8). The highest compressive strength was for PUFs obtained from (CS + GL + H_2_O) + EC and (CS + GL + GLYC) + EC polyols. This is related to the highest functionality of theses polyols, determined by hydroxyl numbers. These polyols provided the best conditions for effective crosslinking because the presence of GLYC resulted in increasing functionalization upon consecutive reactions with GL.

For example, the stress–strain relationship for PUFs obtained from polyol (CS + GL + EC) is shown in the Figure 11. The relationship is typical of rigid PUFs. Initially, the distortion was in line with the compressive strength until 3%. At this point, the PUF lost the ability to transfer the load and the distortion rapidly increased up to 10%, and at this point, the readout of compressive strength of PUF was executed. The PUFs obtained from the composition with 2% water related to the mass of polyol required higher strains forload transfer, which corroborated well with their higher apparent density (compare to Figure 6).

It is worth noticing that after annealing at 150 °C and in some cases at 175 °C, some PUFs showed an increase of compressive strength, presumably due to additional crosslinking upon heating. This was especially well-recognized in PUFs obtained from (CS + GL + H_2_O) + EC and (CS + GL + GLYC) + EC, which had the largest hydroxyl numbers. The PUF obtained from (CS + GL + H_2_O) + EC polyol and 2% water showed 84% and 124% increases of compressive strength after thermal exposure at 150 and 175 °C. In the case of PUFs from the composition with 3% water, a decrease in compressive strength after annealing at 175 °C was observed in relation to the strength measured at 150 °C, but it was often still greater than that determined before the exposure. Nevertheless, in all the cases, compressive strength increased upon annealing, in comparison with the not-annealed PUFs. Generally, two factors influenced the compressive strength changes upon annealing, namely additional crosslinking and degradation, which contributed to increase and decrease the compressive strength, respectively.

The dynamic thermal resistance of PUFs was evaluated by the thermogravimetric method (Figure 12, Table 9). The TG curves clearly demonstrated that PUFs obtained from (CS + GL + H_2_O) + EC and (CS + GL + EC) polyols had the highest thermal resistance. The temperatures of 5% mass loss were 186 °C and 174 °C, or 210 °C and 184 °C, for PUFs obtained from compositions with 2% and 3% water, respectively (Table 9). From these measurements, it has been additionally found that the initial decomposition temperature was lower in case of PUFs obtained with 3% water. The reason for this might be the smaller thickness of the cell walls and the larger pores (see Table 6). Three endothermic peaks were observed at dm/dT vs. temperature curves at 190, 280, and 400 °C. The first peak was attributed to the thermal break of urethane and urea bonds [62], the second one was related to the chitosan ring break [63], and the third to decomposition of polyurethane to amines and carbon dioxide [62]. Decomposition of PUFs was completed at 600 °C. DSC measurements of the obtained PUFs indicated that all samples showed mass loss within 35–110 °C within the first heating/cooling cycle. This endothermic process was due to the presence of TEA and absorbed water. In the second heating cycle, the endothermic process was absent. Therefore, the glass transition temperature could be determined for the PUFs (Figure 12). The glass transition temperature was within 82–148 °C, which allowed the tested materials to be classified as rigid foams (Figure 13). The PUFs obtained from (CS + GL + EC) polyol showed a glass transition temperature beyond this specific range (−40 to 200 °C).

Generally, the chitosan-derived polyols as substrates for the formation of PUFs leads toa new possibility to use them as biodegradable materials. In order to determine the biodegradation of such materials, we have tested their biodegradability. Elemental analyses of the tested polyol and PUF are presented in Table 10. Based on these data, the theoretical oxygen demand (*TOD*) was estimated.

The degree of degradation (Dt) of the studied samples was determined on the basis of BOD after 28 days and the estimated TOD. The 28-day biodegradation test in soil conditions indicated that the degradation percentage of PUF was 52%. This corroborates very well the presence of biodegradable polyol (Table 11). These are very promising results considering that obtained chitosan-based PUFs have good mechanical properties, improved thermal resistance, and a high degree of biodegradation.

The properties of the best PUFs obtained from chitosan can be compared to those of PUFs obtained from polyols based on starch or cellulose (Table 12) [10,11,31,32]. Polyols used to obtain those PUFs were received in similar conditions as described here, namely in aqueous solutions via a reaction with GL and/or alkylene carbonates as hydroxyalkylating agents. Cellulose- and starch-based PUFs had a generally lower apparent density in comparison to these obtained from CS (Table 12). This resulted in a lower compressive strength before thermal workup and a higher mass loss upon thermal exposure. The large mass loss also resulted in a lower compressive strength of the annealed PUFs based on cellulose or starch. Better functional properties of foams based on chitosan also resulted from the comparison of water absorption, which was definitely lower for foams with hydroxyalkylated chitosan units. The PUFs based on CS and obtained from the composition with a lower percentage of water (2%) may find applications as an isolating material because they show a lower thermal conductivity coefficient than those based on cellulose and starch.

## 4. Summary and Conclusions

Four new methods of the synthesis of chitosan-based polyols were elaborated using water-soluble chitosan, glycidol, and ethylene carbonate, with variable environments.The chitosan-derived polyols can be obtained in water in the presence of glycerol or no-solvent conditions. These polyols are suitable to obtain polyurethane foams.The polyurethane foams obtained from these polyols have properties analogous for typical rigid PUFs, except for increased thermal resistance in comparison with classic ones. They can withstand long-term thermal exposure at 175 °C. Additionally, with thermal exposure of the obtained PUFs at 150 °C, the compressive strength of the annealed PUF considerably increased.The chitosan-based polyols and polyurethane foams obtained from them were biodegradable: the polyol was completely biodegraded, while the PUF obtained thereof was 52% biodegradable within 28 days in the soil biodegradation oxygen demand test.Polyurethane foams obtained from polyols based on chitosan converted by the reaction with glycidol and ethylene carbonate in water or in glycerol have useful thermal conductivity, dimensional stability, compressive strength, and low water absorption. Their high thermal resistance renders them as promising candidates to use as thermal insulating materials.

## Data Availability

Not applicable.

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
