# Peer review of "Polyols and Polyurethane Foams Based on Water-Soluble Chitosan"

_polymers, 2023, doi:10.3390/polym15061488_

Round 1
Reviewer 2 Report
Interesting paper on the synthesis of biopolyols from water-soluble chitosan functionalized with chemical modifications for the production of polyurethane foams. The article is well written and full of various types of data and analysis. Aside from a misuse of the term "bio" in the synthesis of such molecules without a true LCA approach, the article sins only on minor errors in language and punctuation. Still, it is worth its publication after minor revision.
Reviewer 3 Report
In this work, authors synthesized biopolyols from water soluble chitosan by functionalization. The products were comprehensive characterized and processed into polyurethane foams further. The obtained foams were characterized with different methods. In general, the manuscript is well organized and the work is interesting. However, there are still some issues to be addressed. A moderate revision is required before its acceptance.
1. One or two sentences are required to present the background and aim of this work in abstract.
2. One scheme to show the experimental procedure is suggested for better understanding of this work to readers.
3. There are too many too old references, which is better to be deleted or replaced with recent articles to show the novelty of this work.
4. Why authors used chitosan in this work should be further clarified with more detailed introduction on the structure, properties and applications of chitosan with necessary supporting articles: Sources, production and commercial applications of fungal chitosan: A review; Recent advancements in applications of chitosan-based biomaterials for skin tissue engineering; etc.
5. Error bar should be added in some of the figures.
6. More details on the raw materials should be provided, such as the purity.
7. Three-line table should be used for a better scientific expression.
8. There are still some typos and grammar issues in the manuscript. Authors should carefully recheck the whole manuscript.
Author Response
You replied to the comments.

Reviewer 4 Report
1. Abstract. It does not highlight the most significant findings. I suggest improving this part.
2. Introduction. This part does not reflect the aim of this study. I suggest adding few sentences why this study was conducted.
3. Why dimensional stability test was chosen to be conducted exactly at 150C temperature? Is there any specific application of the resulting product?
4. According to which method (normative reference) the thermal conductivity was measured? Why the average test temperature was 20C but not 10C?
5. Is the abbreviation "IF" common? Shouldn't it be isocyanate index (II)?
6. The results presented in the whole article show only average values, however, upper and lower limits indicating scattering of the results should be evaluated as well.
7. The whole discussion section misses more in depth analysis of the results and their comparison with other authors works in the same or similar fields.
8. The article is inconsitent because different terms of the same aspects are used. For instance, isocyanate factor - isocyanate index, water uptake - absorption of water or thermal conductivity coefficient - heat conduction coefficient. Please proofread your article and improve this aspect.
9. What kind of water absorption test was conducted? Short-term by immersion or by partial immersion, long-term by immersion or long-term by diffusion?
10. Optical microscopy images in Figure 8 do not show that the foams are closed celled. I do suggest conducting closed cell content determination test to support your statements.
11. What is PUF seasoning? Did authors mean conditioning of the samples?
12. The results discussion lacks more thorough analysis of the results. It is not enough to write only one sentence. the explanations of why one or another thing lead to changes must be explained. I do suggest highly improving this section.
13. Conclusions. Authors analyze few polyols but do not present in the conclusions which is the most suitable according to the results. Also, it would be nice to add the possible application area of such products.
Author Response
You replied to the comments.

Round 2
Reviewer 4 Report
Authors have taken into consideration all my remarks.